



# Soil geochemistry as a driver of soil organic matter composition: insights from a soil chronosequence

Moritz Mainka[1], Laura Summerauer[2], Daniel Wasner[2], Gina Garland[2,3], Marco Griepentrog[2], Asmeret Asefaw Berhe[4], and Sebastian Doetterl[2]

[1]Institute of Landscape and Plant Ecology, Universität Hohenheim, Stuttgart, 70599, Germany
[2]Department of Environmental System Science, ETH Zürich, Zürich, 8092, Switzerland
[3]Agroscope, Soil Quality and Use group, 8046 Zurich, Switzerland
[4]Department of Earth Sciences, Life & Environmental Sciences, University of California, Merced, 95343, USA

*Correspondence to*: Sebastian Doetterl (sdoetterl@usys.ethz.ch)

**Abstract.** A central question in carbon research is how stabilization mechanisms in soil change over time with soil development and how this is reflected in qualitative changes of soil organic matter (SOM). To address this matter, we assessed the influence of soil geochemistry on bulk SOM composition along a soil chronosequence in California, USA spanning 3 million years. This was done by combining data on soil mineralogy and texture from previous studies with additional measurements on total carbon (C), stable isotope values ($\delta^{13}C$ and $\delta^{15}N$), and spectral information derived from Diffuse Reflectance Infrared Fourier-Transform Spectroscopy (DRIFTS). To assess qualitative shifts in bulk SOM, we analysed the peak areas of simple plant-derived (S-POM), complex plant-derived (C-POM), and predominantly microbially derived OM (MOM) and their changes in abundance across soils varying several millennia to millions of years in weathering and soil development. We observed that SOM became increasingly stabilized and microbially-derived (lower C:N ratio, increasing $\delta^{13}C$ and $\delta^{15}N$) as soil weathering progresses. Peak areas of S-POM (i.e. aliphatic root exudates) did not change over time, while peak areas of C-POM (lignin) and MOM (components of microbial cell walls (amides, quinones, and ketones)) increased over time and depth and were closely related to clay content and pedogenic iron oxides. Hence, our study suggests that with progressing soil development, SOM composition co-varies with changes in the mineral matrix. Our study indicates that a discrimination in favour of structurally more complex OM compounds (C-POM, MOM) gains importance as the mineral soil matrix becomes increasingly weathered.

**Keywords:** SOM composition, DRIFTS, [13]C, [15]N, SOM stabilization, soil chronosequence, organo-mineral association.

## 1 Introduction

Soils harbour the largest and most active terrestrial carbon (C) pool on earth (Jobbágy and Jackson, 2000; Lal, 2008). Over the past decades, many of the detailed mechanistic processes determining the fate of soil organic matter (SOM) in soils have been well studied (Schmidt et al., 2011; Lehmann and Kleber, 2015; Kleber et al., 2021). Yet, it remains unclear how these individual processes evolve over time as stabilization mechanisms depend on the soil matrix properties which in turn is subject to constant





changes through soil formation (Bradford et al., 2016; Bailey et al., 2019; Kögel-Knabner and Amelung, 2021). Further,

differences regarding pedogenic (i.e. soil type, mineralogy, soil microbiome) and environmental (i.e. climate, vegetation cover)
properties not only determine the sizes of SOM stocks, but also govern the abundance of diverse biochemical compounds (e.g.
lignin, polysaccharides, lipids) found in the terrestrial C pool (Paul, 2016; Hall et al., 2020). Hence, SOM composition (the

prevalence of certain biochemical compounds) may serve as a proxy for ecosystem properties and soil functioning, as it
responds to land use change (Durrer et al., 2021) and affects C cycling by the amount of energy provided to soil microorganisms
(Nunan et al., 2015; Barré et al., 2016). Furthermore, the peak areas related to biochemical SOM compounds has been
successfully used to parameterize fast and slow-cycling SOM pools (Todd-Brown et al., 2013; Bailey et al., 2018; Laub et al.,
2020; Baldock et al., 2021).

Despite the central role that SOM composition plays in soil C cycling, we lack a thorough understanding of how it is influenced
by the input of organic matter (OM), and by changes of soil C stabilization mechanisms following soil formation (von Lützow
et al., 2008; Doetterl et al., 2018). At initial soil formation stages, the mainly plant-derived input material determines SOM
composition (Khedim et al., 2021). With increasing soil age, soil properties such as soil mineralogy and texture become
increasingly important for the stabilization of accumulating SOM (Chorover et al., 2004; Mikutta et al., 2006; Mikutta et al.,

2009). There is an emerging consensus that certain SOM compounds are preferentially bound to mineral surfaces in organo-
mineral associations (OMA) and are thus highly important C stabilization mechanisms (Kleber et al., 2021). Many studies
reported high affinity of chemically simple carbohydrates to minerals (i.e. low molecular weight compounds from root
exudates or microbial origin) (Kaiser et al., 1997; Calderón et al., 2011; Kleber et al., 2011; Cotrufo et al., 2015; Lavallee et
al., 2019). Conversely, chemically more complex compounds (i.e. lignin or phenols from woody plant parts), formerly

considered to be recalcitrant OM, were found to be less associated with minerals (Spielvogel et al., 2008; Kleber et al., 2011;
Lavallee et al., 2019). However, recent studies reported a preferential binding with iron oxides in the particular case of lignin
(Kramer et al., 2012; Hall et al., 2016).

OMAs are preferentially found in the fine-grained, heavy soil fraction (< 2 μm = clay) which is therefore often regarded as the
most important fraction to explain how SOM persists in soils (Six et al., 2000; Kleber et al., 2021). Recent conceptualizations

argue that the organo-mineral interaction is subject to dynamic changes through mineral weathering (Cotrufo et al., 2013;
Lehmann and Kleber 2015; Kleber et al., 2021). With ongoing soil weathering, primary silicates originating from the parent
material are replaced by newly formed secondary clay minerals (White et al., 1996). The sorptive capacity of clay minerals
thereby decreases as 2:1 layer type silicates (e.g. smectite) are substituted by 1:1 layer type silicates (e.g. kaolinite) (Sposito
et al., 1999). In contrast, iron oxides, particularly nano-sized iron- (Fe) and aluminium-bearing (Al) oxides, gain importance

with increasing soil weathering as they conserve high specific surface areas. Moreover, oxides become positively charged
(protonated) at low pH values allowing for ligand exchange with OM compounds (Kleber et al., 2021). Hence, the potential
of different SOM stabilization mechanisms changes dynamically as a function of mineral weathering (Mikutta et al., 2009).



Given that different SOM compounds have differing binding preferences, mineral weathering is thus likely to affect SOM composition. Therefore, a promising approach to gain further insights into the controls of SOM composition is the observation

of changes in SOM composition along pedogenic gradients.

We sampled from an undisturbed soil chronosequence spanning 3 million years, located under rangeland vegetation in the Mediterranean climatic conditions of California, USA. We used diffuse reflectance infrared Fourier-transform spectroscopy (DRIFTS) to characterize the SOM composition. DRIFTS is a spectroscopic technique commonly used for soil samples and provides spectroscopic information based on the absorbance after an infrared beam hits soil samples. This technique can be

used to semi-quantitatively assess the peak areas of different SOM functional groups (Ellerbrock and Gerke 2004; Parikh et al., 2014; Margenot et al., 2015; Margenot et al., 2017). The spectral information derived from DRIFTS is supported by stable isotope signatures ($\delta^{13}C$ and $\delta^{15}N$ values) and the soil C:N ratio, both of which are strongly associated with SOM dynamics (Dijkstra et al., 2006; Conen et al., 2008; Brunn et al., 2016).

The aim of our study was to assess how bulk SOM composition changed along a gradient of soil mineralogy and texture

associated with soil development. To this end, we used the C:N ratio and the stable isotope signatures $\delta^{15}N$ and $\delta^{13}C$ as proxies for the degree of microbial transformation of the bulk SOM, and the selected peak areas of DRIFTS measurements as proxies for SOM composition. We then combined this information with previously published data on soil mineralogy and texture from the same samples (Doetterl et al., 2018) in order to identify which drivers are most important in explaining shifts in SOM composition with soil development. Based on previous studies, we hypothesized that absorbance peak areas related to simple

plant-derived OM (S-POM) would decrease with increasing soil formation as minerals become increasingly weathered and the formation of OMAs are reduced, hence decreasing the protection of S-POM compounds from microbial degradation and transformation. The peak areas of complex plant-derived OM (C-POM) were expected to increase with soil age due to preferential association with pedogenic iron oxides (Hall et al., 2016; Zhao et al., 2016; Huang et al., 2019). Finally, MOM was expected to strongly increase with soil age since microbial uptake of OM and the subsequent stabilization in soils leads to

an increasing share of microbial-derived SOM (Cotrufo et al., 2013; Cotrufo et al., 2015).

## 2 Material and methods

### 2.1 Site description

The study area is located in the California Central Valley (Merced County, CA, USA), where granitic debris with a minor share of mafic minerals from the Sierra Nevada foothills was deposited on alluvial terraces of the Merced River during

interglacial periods (Harden, 1987; White et al., 1996; White et al., 2005). The alluvial deposits cover a temporal range of 3 million years, with the youngest terrace only 100 years old (Harden, 1987; Doetterl et al., 2018). We assume that the soil age corresponds to the age of the terrace depositions. The climatic conditions are homogeneous across the study region, being semiarid with a mean annual temperature of 16.3 °C and a mean annual precipitation of 315 mm (Reheis et al., 2012). The





vegetation cover is uniform along the chronosequence. On the flat, plateau-like terraces, extensive, unfertilized rangeland
dominates, which is not affected by wildfires. The vegetation consists of annual Mediterranean grassland vegetation, with
common species including *Bromus mollis, Festuca spp.* (grasses), *Erodium spp.* (forbs), and legumes (*Trifolium spp.*) (Jones
and Woodmansee, 1979).

## 2.2 Soil sampling

Soil samples were collected in December 2013 within a diameter of approximately 40 km in the North of Merced County from
1 m³ soil pits (see map in Fig. 1). The terraces are named based on different glacial periods that led to the alluvial deposition
(Harden, 1987). The two youngest terraces are named Post-Modesto (PM) 24II (0.1 kyrs) and Post-Modesto (PM) 22 (3 kyrs).
The intermediate terrace Modesto 2.4 m was deposited between 8 to 30 kyrs ($\emptyset$ 19) ago. The two oldest terraces are Riverbank
deposits (from 260 to 330 kyrs ($\emptyset$ 295) ago) and Chinahat (3000 kyrs old deposits). Per terrace one plot was sampled and three
field replicates were taken per pedogenic horizon (Doetterl et al., 2018). Yet, for the youngest 0.1 kyrs terrace, soils were only
taken to a depth of 30 cm and in the case of the intermediate Modesto 2.4 m terrace, soil samples were taken from three
different soil pits.  To convert the pedogenic horizons to homogeneous depth classes (0-10 and 10-30 cm), weighted arithmetic
means were calculated based on the contribution of each pedogenic horizon to the respective depth class. All soils were
classified as chromic Luvisols except for the 3 kyrs terrace (PM 22) which was classified as a dystric Fluvisol. The total sample
set comprised 30 observations.

## 2.3 Soil analysis

Doetterl et al. (2018) quantified elemental composition, pedogenic oxides, soil pH and cation exchange capacity (CEC), and
soil texture. Elemental composition of the soil samples was determined using inductively coupled plasma-atomic emission
spectrometry (ICP-AES) after borate fusion (Chao and Sanzolone, 1992). Extraction of pedogenic organo-mineral associations
and oxy-hydroxides was done using dithionite-citrate-bicarbonate (DCB) at pH 8 (Mehra and Jackson, 1960). The extracted
pedogenic oxides ($Fe_{DCB}$, $Al_{DCB}$) were then quantified using ICP-AES and serve as a measure of how many Fe- and Al-bearing
phases formed during soil pedogenesis. The degree of mineral weathering along the chronosequence was assessed based on
successive depletion and/or accumulation of geochemical components. For instance, the $Fe_{total}$:$Si_{total}$ ratio indicates how the
mostly pedogenically formed $Fe_{total}$ relates with $Si_{total}$, an element that abounds in the felsic parent material and accumulates
in the semi-arid environment of the study area (White et al., 2005). Hence, the ratio decreases with increasing soil weathering.
Similarly, the Ti:Zr ratio indicates the accumulation of Ti (constituent of many minerals) through mineral weathering relative
to Zr (amount largely determined by the parent material). In addition, the increasing share of organo-mineral associated Fe
($Fe_{DCB}$) and Al ($Al_{DCB}$) was assessed relative to $Fe_{total}$ ($Fe_{total}$:$Fe_{DCB}$) and $Al_{total}$, respectively.

Soil pH was averaged from two measurements, after 30 min and 24 h, in 0.01 M $CaCl_2$ with a glass electrode. The potential
CEC was measured as the $NH_4^+$ exchanged after saturating cation exchange sites with aluminium acetate with 2 M KCl



buffered at pH 7. The share of base saturation ($B_{sat}$) was calculated as the availability of all base cations (K, Na, Mg, Ca) relative to the potential CEC. Soil texture was analysed by Doetterl et al. (2018) with a Mastersizer 2000 laser diffraction particle size analyser (Malvern Instruments). Clay (< 2 µm), silt (2-63 µm), and sand (63 µm – 2 mm) were differentiated based on the WRB classification (WRB 2014). To prevent potential underestimation of clay in silty soils, correction factors were used (Beuselinck et al., 1998; Miller and Schaetzl, 2012).

Soil samples were ground in a ball mill (8000M Mixer/Mill, with a 55 mL tungsten carbide vial, SPEX SamplePrep, LLC, Metuchen, NJ, USA) previous to SOM analysis. To assess the degree of microbial transformation of SOM, we quantified the stable isotope ratios using an elemental combustion system (Costech ECS 4010 CHNSO Analyzer, Costech Analytical Technologies, Valencia, CA, USA) that was interfaced with an isotope ratio mass spectrometer (DELTA V Plus Isotope Ratio Mass Spectrometer, Thermo Fisher Scientific, Inc., Waltham, MA, USA). The stable isotope values were complemented with
total C values taken from Doetterl et al. (2018). Stable isotope values for $\delta^{13}C$ and $\delta^{15}N$ were calculated as the ratio of $^{13}C$ over $^{12}C$ and $^{15}N$ over $^{14}N$ ($R_{sample}$) relative to the Vienna Pee Dee Belemnite (VPDB) standard for $\delta^{13}C$ and relative to atmospheric $N_2$ for $\delta^{15}N$ ($R_{standard}$). Calculation followed Araya et al. (2017):

$$\delta\ [‰] = [(R_{sample} - R_{standard})/R_{standard}]\ x\ 1000 \qquad (1)$$

The spectroscopic analysis was carried out with a Bruker IFS 66v/S Vacuum FT-IR spectrometer (Bruker Biosciences
Corporation, Billerica, MA, USA) to collect diffuse mid-infrared reflectance with a wavenumber range from 4000 to 400 cm$^{-1}$ (2500-25000 nm wavelength). In our analysis, 100 co-added scans were performed on each ground sample and averaged for further analysis. Spectra were normalized with the Standard Normal Variate (SNV) method for scatter correction, and a potassium-bromide (KBr) background for baseline correction. Subsequently, absorbance peak areas were integrated using a local baseline defined as a straight line connecting the absorbances at the upper and lower wavenumber limits of each functional
group (Demyan et al., 2012). Negative peak areas were attributed to noise due to low C stocks and not to excessive signal strength, as they only occured in the subsoil. To account for potential variation in signal strength due to varying C stocks, the peak areas were divided by the respective C stocks. The treatments of the spectra were performed using the OPUS software (Version 7.5.18; Bruker Optik GmbH, Ettlingen, Germany).

We differentiated three functional groups, based on extensive previous research and reviews (Demyan et al., 2012; Vranova
et al., 2013; Parikh et al., 2014; Ryals et al., 2014; Viscarra Rossel et al., 2016; Fissore et al., 2017; Hall et al., 2018): aliphatic C-H bonds, aromatic C=C bonds and C=O bonds (see Fig. 2). These three functional groups are known to be indicative of three groups of SOM compounds with strongly differing chemical properties, and consequently differing availability to microorganisms in soil systems (see Table 1). The wavenumber ranges centered at 2925 cm$^{-1}$ (2976-2898 cm$^{-1}$) and 2850 cm$^{-1}$ (2870-2839 cm$^{-1}$) are attributed to antisymmetric and symmetric aliphatic C-H stretching (Parikh et al., 2014) and represent
simple plant-derived organic compounds (S-POM), e. g. originate from leaf compounds (i.e. waxes) as well as certain organic acids that are secreted by plant roots (i.e. citrate and oxalate) (Vranova et al., 2013). The second wavenumber range, located




at 1512 cm$^{-1}$ (1550-1500 cm$^{-1}$), represents aromatic C=C compounds (Parikh et al., 2014). This functional group is related to lignin-derived, complex plant-derived compounds (C-POM) present in lignin constituents of wooden plant parts and aromatic organic acids of root exudates (Ryals et al. 2014). A large wavenumber range at 1620 cm$^{-1}$ (1660-1580 cm$^{-1}$) represents a

carboxylic C=O stretch of amides, quinones and ketones, with possible shares of aromatic compounds (Parikh et al., 2014). This peak area is interpreted as microbially derived organic compounds (MOM) and jointly represents amides and quinones, which constitute microbial cell walls, and a ketone contribution attributed to long-chained lipids that can also be found in plant materials (Kögel-Knabner and Amelung, 2014). Wavenumber ranges that overlap with signals from mineral compounds, i.e. between 1400 to 400 cm$^{-1}$ (Parikh et al., 2014; Margenot et al., 2017) were excluded from our analysis.

**2.4 Statistical analysis**

Statistical analysis was carried out using R (Version 4.0.1, R Core Team, 2020). Two-way analysis of variance (ANOVA) was carried out to assess differences across soil age (0.1, 3, 19, 295, 3000 kyrs) and soil depth (topsoil: 0-10 and subsoil: 10-30 cm) to identify significant trends of bulk SOM composition parameters. Prior to the ANOVA, residuals were tested for normality with a Shapiro-Wilk test and the homogeneity of variances were checked with a Levene's test. We used Tukey's

Honest Significant Difference (HSD) as a post-hoc test for pairwise comparison of homogeneous subgroups across soil ages. Due to the size of the dataset (n = 30), we controlled for multicollinearity among all variables by calculating variance-inflation factors (VIF) and excluded all variables with VIFs above 5. VIFs quantify the strength of correlation between explanatory variables (Fox and Monette, 1992; Fox and Weisberg, 2019). Out of the variables presented in Table 2 and 3, the Fe$_{total}$:Si$_{total}$ ratio, clay content, Fe$_{DCB}$, and Al$_{DCB}$ were selected as explanatory variables as their inflation factor was below 5. Based on

these variables, linear least squares regressions without interaction terms or non-linear effects were calculated to explain SOM properties (C:N ratio, $\delta^{13}$C, $\delta^{15}$N, S-POM, C-POM, and MOM). The values of the t-statistic were used as a measure of variable importance for each model parameter (Kuhn, 2020). The goodness of model fit was quantified with the R² value. The root mean squared error (RMSE) was quantified as a measure of deviation by a Monte-Carlo cross-validation following a leave-one-group-out principle. Thus, the RMSE was calculated as the difference between predicted values based on a randomly

selected training set (80 % of the observations) in relation to a randomly selected validation set (20 % of the observations) and relative to the sample size. The RMSE as well as the R² values were averaged and respective standard deviations were calculated based on 100 iterations of the cross-validation.

**3 Results**

**3.1 Changes in soil properties and SOM composition along the chronosequence**

With increasing soil age, the soil mineral matrix was progressively weathered. Weathering and leaching of cations led to a reduction of the CEC of 70 % (difference between youngest (0.1 kyrs) and oldest (3000 kyrs) terrace) in 0-10 cm and 85 % in 10-30 cm depth, which was also accompanied by a steady decrease in soil pH along the chronosequence (see Table 2 and 3).





Yet, the gradient observed for CEC was not found in base saturation ($B_{sat}$) values. $B_{sat}$ showed no clear trend with increasing soil age but was consistently higher in the 10-30 cm depth (except for the youngest 0.1 kyrs terrace). We identified a strong

increase by 63 % of $Si_{total}$ in 0-10 cm and by 39 % in the 10-30 cm depth with increasing soil age. $Al_{total}$ and $Fe_{total}$ contents decreased with soil age in both depths. However, this decreasing trend was not displayed in the fractions of DCB-extractable Al ($Al_{DCB}$) and Fe ($Fe_{DCB}$). While $Al_{DCB}$ stagnated along the age gradient, $Fe_{DCB}$ even increased by 54 %. $Fe_{total}$:$Fe_{DCB}$ and $Al_{total}$:$Al_{DCB}$ ratios were highest at intermediate soil development stages. Across all ages and both depths, $Fe_{DCB}$ was the largest fraction of DCB-extracted pedogenic oxides, indicating the high potential of Fe to be quantitatively important for SOM binding

(see Table 2 and 3).

Soil texture showed no clear patterns with soil age, however there were clear differences between soil depths. Except for the youngest terrace, silt and clay fractions were more abundant in 10-30 cm depth, while the proportion of sand was mostly lower relative to the 0-10 cm depth. Overall bulk total C decreased with soil depth, and with soil age in the 0-10 cm depth (bulk C by 81 %). However, the decrease was not linear but peaked at 3 kyrs (see Fig. 3a). Furthermore, the C:N ratio was in general

lower in 10-30 cm depth but did not show a clear trend with increasing soil age (see Fig. 3b). In contrast, $\delta^{13}$C values showed no clear pattern with soil age but increased with soil depth (see Fig. 3c). The $\delta^{15}$N values increased with soil age but showed no significant difference from 0-10 to 10-30 cm depth (see Fig. 3d). Similarly, the peak area of S-POM was lower in the 10-30 cm depth and remained stable with increasing soil age (see Fig. 4a). Conversely, peak areas related to C-POM and MOM increased with increasing age in 0-10 cm depth (C-POM by 94 %; MOM by 90 %) and 10-30 cm depth (C-POM by 87 %;

MOM by 66 %). Moreover, higher peak areas in 10-30 cm suggest a positive trend with soil depth.

### 3.2 Link between soil geochemistry and SOM composition

The DRIFTS peak areas related to S-POM decreased with soil depth while the C-POM and MOM signals rose with increasing soil age and soil depth (see Fig. 4). This raised the question whether there were identifiable soil properties driving the relative changes of S-POM, C-POM, and MOM. Therefore, we used a modelling approach to predict bulk SOM composition using the

soil mineralogy and texture parameters which were not affected by multicollinearity (see Methods). In Table 4, the mean model fit (RMSE and R²) and variable importance coefficients are presented based on the 100 iterations of the cross-validated results of the linear models with different random training and control sets (see Methods). A strong relationship between the variables related to the mineral matrix, soil texture and SOM composition could be observed.

The most important variable explaining the C:N ratio was the $Fe_{total}$:$Si_{total}$ ratio (see Table 4). However, the R² showed high

variability and was overall low. The $\delta^{15}$N values were most influenced by clay content, the $Fe_{total}$:$Si_{total}$ ratio, and $Al_{DCB}$. The model fit was high compared to the C:N ratio model. Similar results were obtained for the $\delta^{13}$C values. Yet, $Fe_{DCB}$ was more important than $Al_{DCB}$. The most important variables explaining the peak areas related to S-POM were $Al_{DCB}$, $Fe_{DCB}$, and the $Fe_{total}$:$Si_{total}$ ratio (see Table 4). Yet, the explained variance was low and variable (0.39 ± 0.33). Conversely, the linear models





explaining the variance of the peak areas linked to C-POM and MOM had high R² values ($R^2$ = 0.78 and 0.57, respectively)
indicating a good fit. The $Fe_{total}:Si_{total}$ ratio followed by clay content contributed most to the model fit of the C-POM response
variable. The clay content followed by $Fe_{DCB}$ were the most important predictors of the MOM peak area.

## 4 Discussion

Our study assessed to what extent SOM composition changes as driven by soil weathering. We found that soil mineralogical
and texture properties were able to explain a large part of variance in SOM composition parameters (Table 4). SOM was
increasingly processed (C:N ratio, $\delta^{13}C$, $\delta^{15}N$) which went along with increasing peak areas related to complex plant-derived
OM (C-POM) and microbial derived OM (MOM), and constant peak areas related to simple plant-derived OM (S-POM) (see
Fig. 5). This suggests that changes in soil mineralogy following weathering may exert control on SOM composition.

### 4.1 Assessing chemical alteration of soil mineralogy, texture, and SOM composition over time and depth

Older soils are increasingly weathered and soil fertility decreases, as indicated by the lower soil pH and CEC (see Table 2 and
3). Further, the increasing dominance of Si with ongoing mineral weathering indicates increasing amounts of low reactive
clay-sized silicates, i.e. kaolinite, which offer less binding sites to SOM (White et al., 2005). At the same time, lower total C
in strongly weathered soils were accompanied by increasing amounts of pedogenic iron oxides, i.e. $Fe_{DCB}$. Similar effects were
observed in previous soil chronosequence studies, highlighting the importance of iron-bearing mineral phases for SOM
stabilization in strongly weathered soils (Chorover et al., 2004; Mikutta et al., 2009). In this study, the increases in pedogenic
iron oxides co-varied with proxies that indicate increasingly stabilized and microbially transformed SOM (decreasing C:N
ratio, increasing $\delta^{13}C$ and $\delta^{15}N$ values) and a shift towards a C-limited soil environment (Coyle et al., 2009; Cotrufo et al.,
2021). The stronger microbial-derived origin of SOM is reflected in the increasing $\delta^{13}C$ and $\delta^{15}N$ values that develop due to
isotopic discrimination of microorganisms against heavier $^{13}C$ and $^{15}N$ (Dijkstra et al., 2006; Dijkstra et al., 2008; Kramer et
al., 2017).

Soil fertility and SOM stabilization capacity of soils decreased along the chronosequence, thus leading to lower total C stocks
on old terraces and stronger signs of microbially transformed SOM following reduced C input (Doetterl et al., 2018).
Additionally, we observed changes in the peak areas of different functional groups of different origins and complexity
regarding their chemical structures. While peak areas of simple plant-derived OM (S-POM) were less affected over time but
decreased with depth, peak areas of microbial-derived (MOM) and complex plant-derived OM (C-POM) increased over time
and with depth (see Fig. 4). We related S-POM to aliphatic compounds from litter and/or root exudates, C-POM compounds
to aromatic structures of aboveground woody debris and/or aromatic root constituents, and MOM to long-chained lipids of
predominantly microbial origin (see Table 1).





In contrast to our expectation, S-POM did not decrease along the chronosequence as the vegetation provided steady supply by above- (light fraction of litter compounds) and belowground input (low-molecular weight root exudates, Nardi et al., 2005)

particularly in the topsoil layer. The decreasing peak areas of S-POM with depth might be related to a lower stabilization potential of aliphatic organic acids with increasing soil depth (Vranova et al., 2013). Yet, it is noteworthy that below the main rooting zone (> 16 cm depth) increases of aliphatic compounds have been observed in similar grassland soils which was explained by increasing stabilization on mineral surfaces (Feng et al., 2007). The latter is further supported by the high affinity of S-POM, in particular citrate and oxalate, to bind with Fe and Al oxides ($Fe_{DCB}$ increased with soil age) in acidic soils

(Clarholm et al., 2015). The relative increase of MOM peak areas in parallel to the constant S-POM peak areas along the chronosequence could indicate that an increasing fraction of microbially processed and transformed OM gains importance in bulk SOM with soil age (Feng et al., 2007). Previous studies showed a strong correlation between the MOM peak area and organo-mineral associations (Kaiser et al., 2007; Demyan et al., 2012; Kaiser et al., 2012). Thus, the observed increases in our study hint towards an increasing importance of OMAs with progressing soil weathering. C-POM signals, interpreted as lignin-

compounds from woody plant parts, behaved similarly to MOM and increased with soil age and depth. In grassland soils, increases of signals related to lignin compounds with depth were explained by the increasing dominance of root-derived OM input in subsoils (Feng et al., 2007). Yet, increases in lignin compounds were also observed following a preferential association with iron oxides which is supported by the increase of pedogenic $Fe_{DCB}$ with soil age (Hall et al., 2016; Zhao et al., 2016) as well as kaolinite which is supported by the increasing clay contents in our samples with soil depth (Li et al., 2019).

**4.2 Mineralogical control of SOM dynamics over geological time**

As indicated above, the changes in the degree of decomposition (as indicated by the C:N ratio, $\delta^{13}C$, and $\delta^{15}N$) and SOM composition (peak areas of S-POM, MOM, and C-POM) must be contemplated in the light of simultaneously occurring changes of the mineral matrix along the chronosequence. Therefore, in this section we discuss to which extent the relationship between soil mineralogy, texture, and SOM (de-)composition is reflected in linear models and how these findings can be

embedded in our current understanding of SOM dynamics.

The properties related to SOM decomposition showed differences in the model fit (see Table 4). The C:N ratio showed the lowest model fit of all SOM composition properties. Low and variable R² values highlighted that the mineral matrix did not strongly affect the C:N ratio of bulk SOM even though SOM was progressively decomposed with increasing soil age. The only variable that contributed significantly to the explanation of the C:N ratio was the $Fe_{total}:Si_{total}$ ratio. The decrease of the

$Fe_{total}:Si_{total}$ ratio with soil age indicated that during soil weathering processes, low reactive silicates, i.e. quartz, or 1:1 layer type clay minerals (kaolinite), abound (White et al., 2005). The second most important predictor variable, $Fe_{DCB}$, reflects an increasing importance of pedogenic iron oxides in SOM stabilization as these compounds retain very high reactive surface





areas (Kleber et al., 2021). Our findings show that the properties of the mineral matrix play a subordinate role in shaping the C:N ratio which is more strongly driven by the increasing microbial origin of the SOM (Sollins et al., 2009).

In contrast to the C:N ratio, linear models with $\delta^{13}C$ and $\delta^{15}N$ values as response variables showed a good model fit of $R^2 >$ 0.60 (see Table 4). The $\delta^{13}C$ values were strongly related to $Fe_{DCB}$, $Fe_{total}$:$Si_{total}$, and clay, while $\delta^{15}N$ values were explained by clay, $Fe_{total}$:$Si_{total}$, and $Al_{DCB}$. In both cases, the high importance of clay content as predictor supports empirical findings that SOM with a higher share of microbial-derived OM (high $\delta^{15}N$; high MOM signal) and increasingly decomposed OM (low C:N ratio; high $\delta^{15}N$, less negative $\delta^{13}C$) is predominantly found within the heavy (clay) fraction of soils (Conen et al., 2008; Sollins

et al., 2009; Clemente et al., 2011; Lawrence et al., 2015). It has been previously demonstrated that SOM bound to iron oxides was enriched in $\delta^{13}C$ and exhibited more MOM (Zhao et al., 2016). Likely the imprint of $\delta^{13}C$ enrichment with increasing microbial transformation of the SOM manifested itself more clearly in bulk SOM as the overall C stocks decreased in the oldest soils (Yang et al., 2015).

Similar to the models explaining SOM decomposition variables, the linear models on DRIFTS peak areas had varying model
fits (see Table 4). The model for S-POM had a lower model fit ($R^2 = 0.39$) compared to the models explaining C-POM and MOM ($R^2 = 0.78$ and $0.57$, respectively). In all models, the clay content was a key explanatory variable which hints at different behaviours in top- and subsoils as consistently higher shares of clay were found in the 10-30 cm depth (see Table 3). The fact that S-POM was less influenced by the $Fe_{total}$:$Si_{total}$ ratio suggests that the contribution of S-POM to OM is less related to weathering-induced changes of the mineral matrix, but rather determined by above- and belowground vegetation input, i.e.
through litter components or root exudates (Vranova et al., 2013; Mueller et al., 2013). Lower signals of S-POM in subsoils compared to topsoils (see Fig. 4) indicate that the mineral stabilization of S-POM compounds in subsoils is lower or that S-POM compounds are less abundant at these depths (Vranova et al., 2013). Still, the relative variable importance of $Al_{DCB}$ and $Fe_{DCB}$ to explain S-POM indicates that certain compounds, likely aliphatic organic acids derived from root exudates, are preferentially attached to minerals (Clarholm et al., 2015).

In the case of C-POM, the decreasing $Fe_{total}$:$Si_{total}$ ratio with increasing soil age was the most important predictor. This supports the notion that functional groups that are considered chemically more complex (i.e. through an aromatic ring structure) are increasingly stabilized relative to other functional groups, as soils become more weathered, and as overall SOM stocks deplete along with reduced protection through reactive minerals in strongly weathered soils (Angst et al., 2018). Lignin compounds represented by C-POM were shown to be better stabilized due to attachment on mineral surfaces, such as calcite (Grünewald
et al., 2006), kaolinite (Li et al., 2019), as well as Fe oxides (Kramer et al., 2012; Hall et al., 2016; Zhao et al., 2016; Huang et al., 2019). Yet, the coefficients of variable importance of the C-POM model did not show a particular importance of $Fe_{DCB}$ and $Al_{DCB}$ (see Table 4). Potentially the more linear decrease of the $Fe_{total}$:$Si_{total}$ ratio might mask less linear increases of $Fe_{DCB}$ or the varying behaviour of $Al_{DCB}$ (see Table 2 and 3). Instead, clay content was the second most important predictor, and it was previously shown that kaolinite accumulates in the mineral matrix with increasing soil age and depth along the present
chronosequence (White et al., 2005). Hence, the mineral stabilization of C-POM in heavy soil fractions might be more strongly linked to the presence of clay minerals than iron oxides (Li et al., 2019; Grünewald et al., 2006).

The most important predictors to explain the peak areas related to MOM were clay content and $Fe_{DCB}$ (see Table 4). These predictors highlight the similarity of the MOM peak area with variables like $\delta^{13}C$ that are related to SOM decomposition. Both properties reflect the increasingly microbial-derived imprint in the bulk SOM. The clay fraction is known to harbour the vast

majority of soil microorganisms and to stimulate their activity (i.e. Stotzky and Rem, 1966; Sollins et al., 2009). Our study is in line with these findings as MOM signals increased with increasing clay content. However, the MOM peak also showed a strong link to $Fe_{DCB}$ suggesting a link between increasing importance of pedogenic iron oxides and MOM along the soil chronosequence. This might be caused by the increasing importance of metal surfaces in stabilizing bulk SOM, but also soil microorganisms or microbial necromass in particular (Kaiser et al., 1997, Zhao et al., 2016). Yet, these interpretations are more

speculative due to the heterogeneity of soil microorganisms and the vast range of possible interactions requires further investigation.

**5 Conclusion**

The present observational study provides new information on qualitative changes of the SOM composition in relation to soil geochemical dynamics driven by mineral weathering from a soil chronosequence covering a gradient of 3 million years. As

SOM stocks decreased, we observed an increasingly microbially transformed bulk SOM (lower C:N ratio, higher $\delta^{13}C$ and $\delta^{15}N$ values) with progressing soil development. Simultaneously, the spectroscopic fingerprint of SOM (reflected in the peak areas related to S-POM, C-POM, and MOM) shifted, exhibiting increasing peak areas of structurally more complex compounds (C-POM and MOM). We relate these qualitative changes in SOM composition to changes in the soil geochemistry induced by soil formation and suggest that changes in soil texture (clay content) and an increasingly kaolinite- and iron oxide-rich, strongly

weathered mineral matrix discriminates in favour of structurally more complex SOM compounds (C-POM and MOM).

**Author contributions**

AAB and SD designed the experiments. MM performed the measurements, analysed the data and wrote the manuscript. DW, GG, MG, and LS reviewed and edited the manuscript. LS created the map (Fig. 1). All authors contributed to writing the paper (Lead author: MM).

**Competing interests**

The authors declare that they have no conflict of interest.




**Acknowledgements**

This study was carried out with the financial support of the Bavaria California Technology Center (BaCaTeC) (Project 22, 2016-1). We thank the working group of Asmeret A. Berhe for logistic help and support in the laboratory.

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

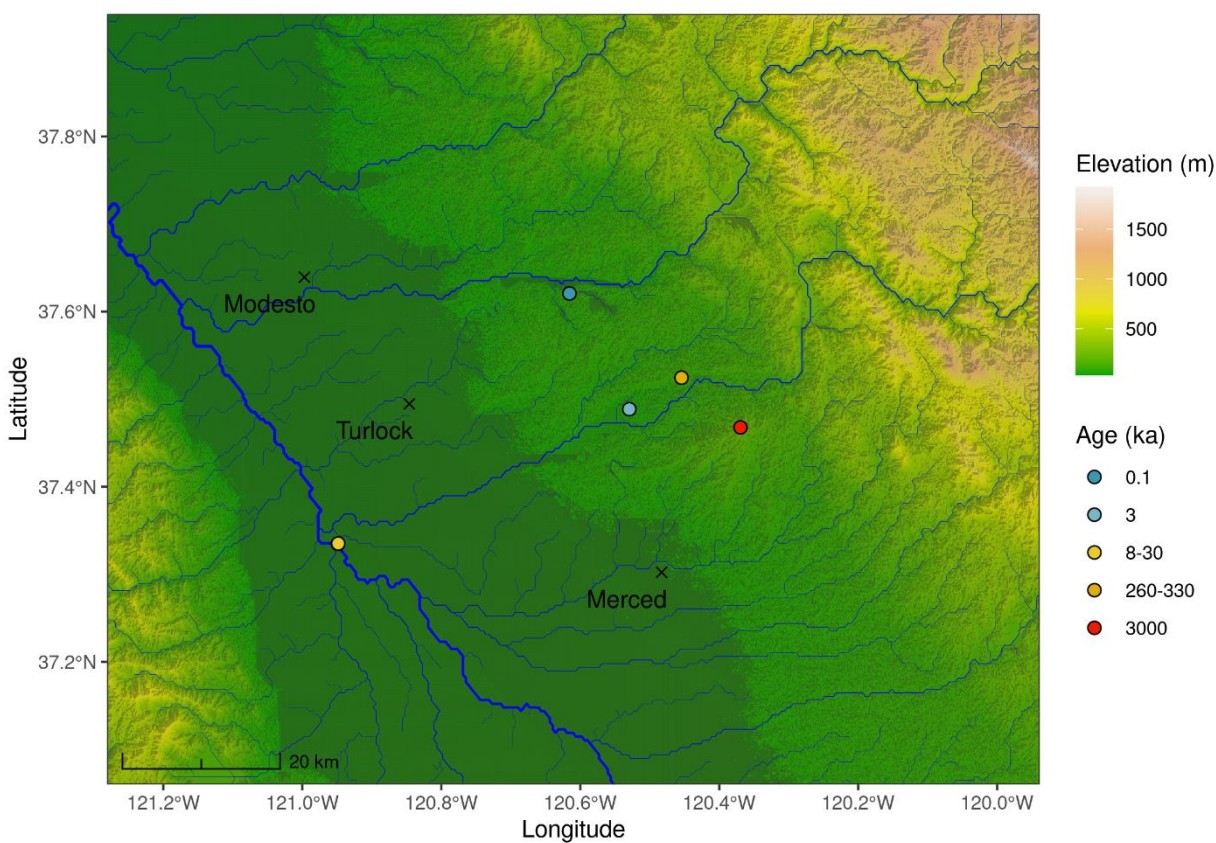

**Figure 1: Map of the study area based on a digital elevation model (DEM) retrieved from Shuttle Radar Topography Mission**
**(SRTM, resolution ~ 30 m). River hydrology data was taken from Lehner et al. (2008).**



**Table 1: Overview of wavenumber ranges used to calculate absorbance peak areas from DRIFTS, the respective functional groups related to specific biochemical properties of SOM compounds and compiled information on the origin (Demyan et al., 2012; Parikh et al., 2014; Hall et al., 2018) and the respective SOM properties (C:N ratio, $\delta^{13}C$, and $\delta^{15}N$ values) of each functional group (Benner et al., 1987; Austin and Vitousek, 1998; Dijkstra et al., 2006; Xu et al., 2010; Vranova et al., 2013).**

| SOM type | Wavenumber center (range) [cm$^{-1}$] | Functional group | Origin | SOM properties |
|---|---|---|---|---|
| **S-POM** | 2925 (2976-2898) + 2850 (2870-2839) | aliphatic C-H stretch | constituent of litter, component of many root exudates (i.e. citrate, oxalate) | intermediate C:N ratio, $\delta^{13}C$ and $\delta^{15}N$ depleted compared to soil but higher compared to C-POM |
| **C-POM** | 1525 (1550-1500) | aromatic C=C stretch | mostly lignin, related to wooden plant parts, also found in root constituents and aromatic root exudates (i.e. phthalic or vanillic) | high C:N ratio, $\delta^{13}C$ and $\delta^{15}N$ depleted |
| **MOM** | 1620 (1660-1580) | amide, quinone, ketone C=O stretch, aromatic C=C and/or carboxylate C-O stretch | constituents of microbial cell walls (amides and quinones), and long-chained lipids (ketones) | low C:N ratio, $\delta^{13}C$ and $\delta^{15}N$ enriched |






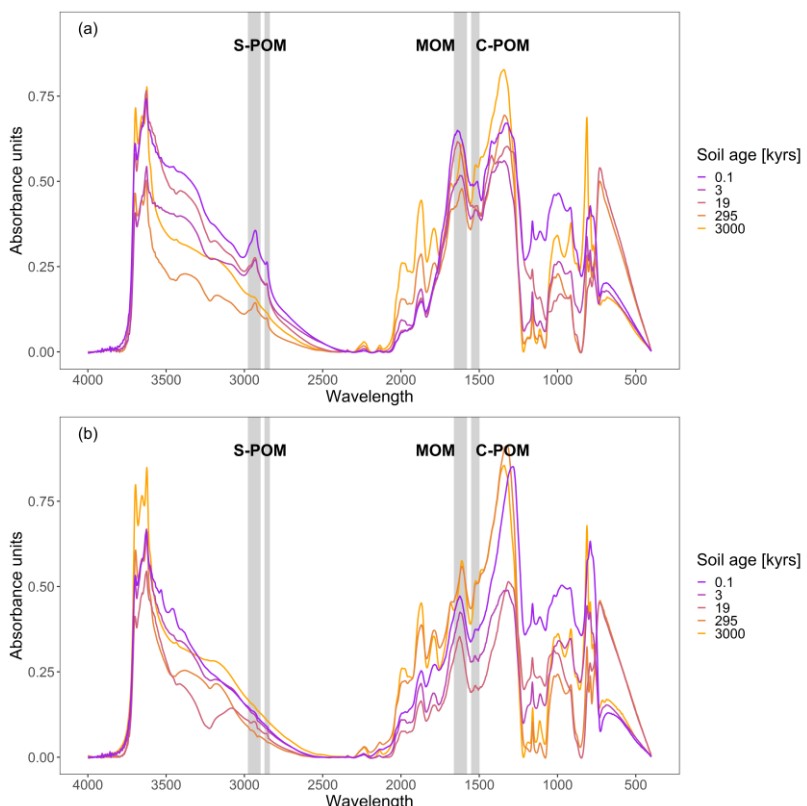

**Figure 2: Baseline corrected DRIFTS spectra of each soil age for 0-10 cm (2a) and 10-30 cm (2b) soil depth. The wavenumber ranges across which absorbance peak areas were integrated for functional group assignment are indicated with grey bars in the background. The two wavenumber ranges around 2900 cm⁻¹ stand for simple plant-derived OM (S-POM), the one around 1620 cm⁻¹ for microbially-associated OM (MOM), and the one at 1525 cm⁻¹ for complex plant-derived OM (C-POM).**





**Table 2: Overview of soil mineralogy and texture changes with increasing soil age in 0-10 and 10-30 cm depth (n = 30). The mineralogy was measured using inductively coupled plasma-atomic emission spectrometry (ICP-AES). Texture was quantified with 565 a laser diffraction particle size analyser. Pedogenic oxides were extracted in duplicates with dithionite-citrate-bicarbonate (DCB).**

| Soil depth [cm] | Soil age [kyrs] | $Al_{total}$ [g/kg] | $Fe_{total}$ [g/kg] | $Si_{total}$ [g/kg] | $Fe_{total}:Si_{total}$ | $Al_{DCB}$ [g/kg] | $Fe_{DCB}$ [g/kg] | $Fe_{total}:Fe_{DCB}$ | $Al_{total}:Al_{DCB}$ | Ti:Zr | Clay [%] | Silt [%] | Sand [%] | CEC [meq/100g] | $B_{sat}$ [%] | pH |
|---|---|---|---|---|---|---|---|---|---|---|---|---|---|---|---|---|
| 0-10 | 0.1 | 82.7 | 36.5 | 241. | 0.15 | 1.3 | 6.2 | 5.9 | 65.1 | 21.6 | 15.2 | 58.4 | 26.4 | 15.2 | 86.6 | 6.8 |
| 0-10 | 3 | 65.0 | 35.3 | 293. | 0.12 | 1.0 | 9.5 | 3.7 | 66.2 | 19.6 | 9.1 | 62.6 | 28.2 | 13.4 | 109. | 6.6 |
| 0-10 | 19 | 81.1 | 38.4 | 267. | 0.14 | 1.1 | 6.6 | 7.6 | 86.9 | 26.8 | 11.2 | 64.3 | 24.5 | 24.7 | 81.1 | 5.9 |
| 0-10 | 295 | 63.9 | 18.2 | 338. | 0.05 | 0.7 | 6.2 | 2.9 | 88.5 | 13.8 | 13.7 | 50.3 | 36.0 | 4.6 | 103. | 6.1 |
| 0-10 | 3000 | 29.3 | 23.1 | 393. | 0.06 | 1.3 | 13.6 | 1.7 | 22.3 | 10.8 | 11.1 | 68.1 | 20.8 | 4.5 | 58.3 | 4.7 |
| 10- | 0.1 | 92.2 | 37.1 | 280. | 0.13 | 1.0 | 6.1 | 6.1 | 102. | 18.0 | 4.9 | 63.1 | 32.0 | 14.2 | 74.3 | 6.8 |
| 10- | 3 | 73.4 | 39.6 | 306. | 0.13 | 0.9 | 10.4 | 3.8 | 80.3 | 14.2 | 12.9 | 69.2 | 17.9 | 9.9 | 116. | 6.2 |
| 10- | 19 | 87.1 | 43.7 | 273. | 0.16 | 1.3 | 7.2 | 6.4 | 84.7 | 24.4 | 17.4 | 69.3 | 13.4 | 20.8 | 95.5 | 6.0 |
| 10- | 295 | 64.9 | 18.4 | 348. | 0.05 | 0.6 | 6.5 | 2.8 | 106. | 14.6 | 19.0 | 53.3 | 27.7 | 2.1 | 131. | 5.8 |
| 10- | 3000 | 36.7 | 25.3 | 389. | 0.06 | 1.4 | 13.5 | 1.9 | 26.2 | 10.1 | 18.3 | 69.2 | 12.5 | 2.1 | 85.3 | 4.3 |



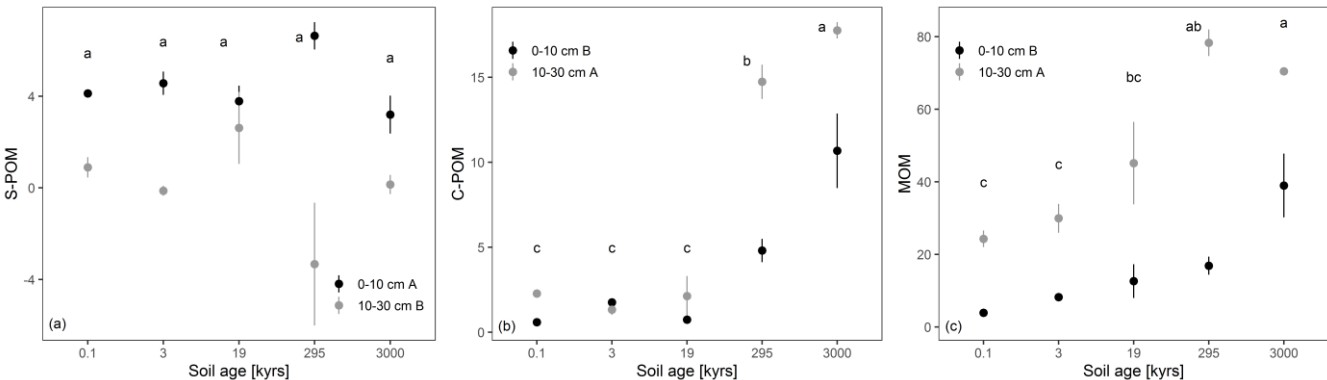

**Figure 3: Mean ± standard deviation of site triplicates of total C (3a), the C:N ratio (3b), δ¹³C (3c), and δ¹⁵N (3d) for 0-10 cm (black) and 10-30 cm soil depth (grey). Significant differences between the soil ages calculated with the Tukey-HSD test are indicated with letters (p < 0.05). Differences across soil age are indicated with lowercase letters, differences across soil depth in capital letters.**


**Figure 4: Mean ± standard deviation of site triplicates of the DRIFTS absorbance peak areas corresponding to simple plant-derived OM (S-POM; 4a), complex plant-derived OM (C-POM; 4b), and mainly microbial-derived OM (MOM; 4c) for 0-10 cm (black) and 10-30 cm soil depth (grey). Homogeneous subgroups (p < 0.05) across soil age are indicated with letters. Significant differences**





between soil depth are indicated with letters behind the depth classes in the legend. The subgroups were identified by pairwise comparison (Tukey-HSD) following a two-way ANOVA (n = 30).

**Table 3:** Overview of the linear regression models describing SOM composition (C:N ratio, $\delta^{15}N$ and $\delta^{13}C$), and peak areas of functional groups related to simple plant-derived OM (S-POM), complex plant-derived OM (C-POM) and mainly microbially-derived OM (MOM) based on n = 30 observations. Explanatory variables related to the mineral matrix and soil texture were selected 580 ($Al_{DCB}$, $Fe_{DCB}$, $Fe_{total}$:$Si_{total}$ ratio, clay content). Model fit parameters (root mean squared error (RMSE) and $R^2$) with standard deviations were computed following 100 iterations of a Monte-Carlo cross-validation. Higher coefficients of variable importance indicate a higher importance of the variable to explain the respective response variable.

| | C:N ratio | $\delta^{15}N$ | $\delta^{13}C$ | S-POM | C-POM | MOM |
|---|---|---|---|---|---|---|
| **RMSE** | 2.0 ± 0.5 | 1.2 ± 0.5 | 0.6 ± 0.2 | 2.9 ± 1.0 | 3.2 ± 0.7 | 20.3 ± 5.7 |
| **R²** | 0.32 ± 0.27 | 0.64 ± 0.24 | 0.61 ± 0.32 | 0.39 ± 0.33 | 0.78 ± 0.10 | 0.57 ± 0.27 |
| **Coefficients of variable importance** | | | | | | |
| **$Al_{DCB}$** | 1.3 | 1.3 | 3.3 | 2.3 | 1.0 | 1.1 |
| **$Fe_{DCB}$** | 1.8 | 0.6 | 5.2 | 2.1 | 1.0 | 1.6 |
| **$Fe_{total}$:$Si_{total}$** | 3.2 | 1.4 | 5.0 | 1.4 | 5.0 | 0.8 |
| **Clay** | 0.4 | 1.5 | 4.1 | 2.2 | 2.8 | 3.7 |

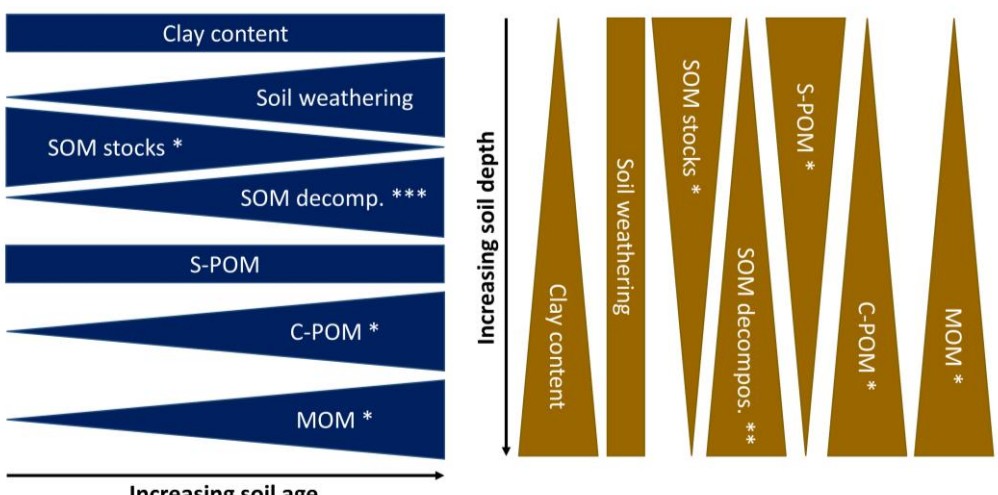

**Figure 5:** Conceptual model explaining the influence of soil development with increasing soil age and depth on mineralogy and texture based on our findings. These properties (i.e. $Fe_{total}$:$Si_{total}$, $Fe_{DCB}$) may then drive the dynamics of SOM stocks, properties, and composition. SOM stocks reflect the total C stock, SOM decomposition proxies include $\delta^{13}C$, $\delta^{15}N$, and the C:N ratio, and the signals of the selected DRIFTS peak areas are represented individually. Duplicates of soil weathering proxies and clay content were not tested. SOM decomposition proxies and peak areas were tested by a two-way ANOVA (n = 30) with Tukey-HSD as post-hoc test (p 590 < 0.05): * showed significant trends. ** significant trends for C:N and $\delta^{15}N$ but not for $\delta^{13}C$. *** significant trends for C:N and $\delta^{13}C$ but not for $\delta^{15}N$. The outcomes of the ANOVA are visualized in Fig. 3 and 4, the values of soil geochemistry in Table 2.