# Peer review of "Soil geochemistry as a driver of soil organic matter composition: insights from a soil chronosequence"

_Biogeosciences, 2021_

## Author Comment (AC1)

**Point-by-point response to Referee#1 comments:**

Dear Mr. Dupla,

We would like to thank you for the thorough evaluation of our manuscript "Soil geochemistry as a driver of soil organic matter composition: insights from a soil chronosequence" by Mainka et al. 2021 (bg-2021-295; https://doi.org/10.5194/bg-2021-295). We are very pleased about your positive assessment of our work and recognition of its relevance. Your comments helped us to significantly improve our manuscript and we want to sincerely thank you for the constructive and valuable insights. We have addressed all comments and suggestions to the best of our ability. Please find below a point-by-point response to all the concerns raised and how we will address them. Reviewer original comments are highlighted in grey. New text to be added or modified in the manuscript can be found at the end of each response.

We hope you find our response and changes to the manuscript satisfying and we are looking forward to hearing from you.

Yours sincerely,

The authors

**Reviewer#1 Comment#1 (l. 58):**

remove thereby which is misleading with the previous sentence

**Our response:** We thank the reviewer for this comment. The misleading word was deleted.

"The sorptive capacity of clay minerals  decreases as 2:1 layer type silicates (e.g. smectite) are substituted by 1:1 layer type silicates (e.g. kaolinite) (Sposito et al., 1999)."

**Reviewer#1 Comment#1 (l. 61):**

oxide do not "become" positively charged at low pH values, they are positively charged on the whole pH range of almost all soils (check pzc values). If you want to open the pandora box of variable charges, it is difficult to speak about the protonation of surfaces without saying that several OM functional groups too protonates when pH becomes acidic. Furthermore, if you want to maintain this sentence about acidic soils, then you could more explicitly mention that soil acidification is a key process behind soil weathering.

**Our response:** Indeed, the high complexity of variable charges of minerals AND organic matter compounds was described too simplistically. Therefore, we will modify the sentence to avoid an in-depth discussion of this topic (as it is not the main focus of our study) but keep the link between soil acidification and soil weathering.

 As soil acidification progresses with soil weathering, the importance of these oxides to stabilize OM compounds becomes more pronounced (Kleber et al., 2021).

**Reviewer#1 Comment#3 (l. 79):**

this section contradicts what you say from line 47 onwards. Rephrasing either the upper section (lines 47-52) of this one (lines 79-85) might help

**Our response:** We thank the referee for making us aware of this issue. We will rephrase the hypotheses statements in lines 79-85 as suggested below. With these changes it should be clearer why we expected S-POM to decrease with mineral weathering on the presented time span. The high dependency of S-POM on mineral protection leads to decreases with soil age since less binding sites are available in highly weathered soils.

"Based on previous studies, we hypothesized that absorbance peak areas related to simple plant-derived OM (S-POM) would decrease with increasing soil formation as binding sites for  the formation of OMAs are reduced. Consequently,  the protection of S-POM compounds from microbial degradation and transformation decreases."

**Reviewer#1 Comment#4 (Discussion):**

overall excellent. However, you did not notice any significant decrease in base saturation along your chrono-sequence which contradicts general description of soil weathering sequences. This aspect is extremely interesting and should be discussed.

**Our response:** We thank the reviewer for this positive feedback regarding our discussion section. We acknowledge the issue raised by the reviewer and would kindly like to state that the behaviour of pedogenic (DCB-extractable) iron oxides along the chronosequence behaves consistently with the base saturation values and might offset the decreases in CEC and pH that occurred with increasing weathering. We decided to include the following sentence in the discussion section (in 4.1, l. 231-232):

"At the same time, lower total C in strongly weathered soils were accompanied by increasing amounts of pedogenic iron oxides, i.e. $Fe_{DCB}$ that likely contributed to the unaltered base saturation values."

**Reviewer#1 Comment#5 (non-binding suggestion):**

your discussion sticks very closely to the parameters. I was expecting your paper to zoom out at some point in order to 1) discuss how the climate and geomorphological changes that happened in your 3 million-year sequence may have impacted your results 2) discuss the general impact of your findings on our understanding of soil weathering, 2) outline the limits of your study and what should be done to go further

**Our response:** We thank the author for this comment. While we do understand the importance of zooming out to assess the broader implications of our research, our aim was to keep the scope of the current paper narrow and to strictly focus on the impact of mineralogical changes on soil organic matter composition and decomposition proxies across this particular chronosequence. As different processes occur under different soil types and across different timescales, our results are not necessarily applicable or thus generalizable across a larger

scale. However, to add a final, summarizing sentence, we added the following at the end of the conclusion section:

"Our study shows that soil mineralogy plays an important role in shaping SOM composition during soil weathering across large timescales. We therefore recommend further studies to assess these trends in contrasting soil mineralogies in order to gain a better understanding across larger geographical areas."

---

## Author Comment (AC2)

**Point-by-point response to anonymous Referee#2 comments**

Dear Referee#2,

We would like to thank you for your time and thorough evaluation of our manuscript "Soil geochemistry as a driver of soil organic matter composition: insights from a soil chronosequence" (https://doi.org/10.5194/bg-2021-295). We are very pleased that you positively assessed our work and recognized its relevance. Your comments helped us to significantly improve our manuscript and we want to sincerely thank you for the constructive and valuable insights. We are convinced that we were able to address the main concern of the reviewer and especially improve the description of our results to make our approach and measurements more accessible to the readers. We have responded to all comments and suggestions to the best of our ability. Please find below a point-by-point response to all the concerns raised and how we addressed them. Reviewer original comments are highlighted in grey. New text to be added or modified in the manuscript can be found at the end of each response.

We hope you find our response and changes to the manuscript satisfying and we are looking forward to hearing from you.

Yours sincerely,

The authors

**Reviewer#2 Comment#1 (l. 36):**

You mention "responds to land use change" but I'm not too sure whether it is that relevant to mention it here, because land use change will have an impact on SOC stabilization mechanisms on a much shorter time period (scale of 10-100 years), whereas you are looking to a timescales exceeding 1000s of years.

**Our response:** We thank the reviewer for this important comment. We suggest the following phrase since indeed it is not relevant to our research at millenial timescales:

"Hence, SOM composition (the prevalence of certain biochemical compounds) may serve as a proxy for ecosystem properties and soil functioning, as it  affects C cycling by the amount of energy provided to soil microorganisms (Nunan et al., 2015; Barré et al., 2016)."

**Reviewer#2 Comment#2 (l. 40-65):**

I like the idea of the interaction between physical and chemical stabilization mechanisms and associated changes over time. But as regards the physical protection, I can see that the focus is mainly on sorption / binding of OM to mineral surfaces (and as such also the importance of clay % and type of clay in this context). However, I was wondering whether soil aggregate formation (macros and micros) shouldn't be considered as well / more explicitly here? (or is this not the right time scale. But if so, I guess you should neither mention 'land use change' – see my comment just above this one).

**Our response:** We would like to thank the reviewer for commenting on this important aspect in SOC stabilization. We fully agree that soil aggregation is a very important factor, and

influences SOC stabilization at millenial timescales since the aggregation capacity of soils increases with soil weathering (Wei et al., 2016). Hence, we will explicitly mention soil aggregation behaviour along a weathering gradient in our considerations on SOC stabilization mechanisms as proposed below. Still, the main focus on our analysis remains on the geochemical aspect. We will remove land use change as an infuencing factor on C stabilization, as it is not relevant on this timescale (see above).

References:

Wei, Y., Wu, X., Xia, J., Shen, X., Cai, C.: Variation of Soil Aggregation along the Weathering Gradient: Comparison of Grain Size Distribution under Different Disruptive Forces, PLoS ONE, 11: e0160960. doi:10.1371/journal.pone.0160960, 2016.

L. 43-45: "With increasing soil age, soil properties such as soil mineralogy,  texture, and soil aggregation become increasingly important for the stabilization of accumulating SOM (Chorover et al., 2004; Mikutta et al., 2006; Mikutta et al., 2009; Wei et al., 2016). A comparative study of soil aggregation in relation to soil weathering found micro- and macroaggregates to become more stable with increasing weathering (Wei et al., 2016)."

**Reviewer#2 Comment#3 (l. 66-69 & 75-85):**

I think that the information given in these sections mainly belongs to the Material & Methods section. Hence, I would like to suggest to rewrite the end of the introduction section so that you end with making the problem statement clear followed by your main objective(s), i.e. the contribution of present research in addressing this particular problem statement.

**Our response:** We would like to thank the reviewer for the suggestion. We fully agree, that this part of the introduction contained too technical information that is repeated in the Materials and Methods section. Hence, we suggest the modifications indicated below to improve our manuscript:

~~We sampled from an undisturbed soil chronosequence spanning 3 million years, located under rangeland vegetation in the Mediterranean climatic conditions of California, USA. We used diffuse reflectance infrared Fourier transform spectroscopy (DRIFTS) to characterize the SOM composition. DRIFTS is a spectroscopic technique commonly used for soil samples and provides spectroscopic information based on the absorbance after an infrared beam hits soil samples.~~

The aim of our study was to assess how bulk SOM composition changed along an undisturbed soil chronosequence spanning 3 million years, located under rangeland vegetation in the Mediterranean climatic conditions of California, USA. To this end, we used the C:N ratio and the stable isotope signatures $\delta15N$ and $\delta13C$ as proxies for the degree of microbial transformation of the bulk SOM, and the selected peak areas of diffuse reflectance infrared Fourier-transform spectroscopy (DRIFTS) measurements as proxies for SOM composition. ~~This technique can be used to semi-quantitatively assess the peak areas of different SOM functional groups (Ellerbrock and Gerke, 2004; Parikh et al., 2014; Margenot et al., 2015; Margenot et al., 2017). The spectral information derived from DRIFTS is supported by stable isotope signatures (δ13C and δ15N values) and the soil C:N ratio, both of which are strongly associated with SOM dynamics (Dijkstra et al., 2006; Conen et al., 2008; Brunn et al., 2016).~~ We then combined this information with previously published data on soil mineralogy and texture from the same samples (Doetterl et al., 2018) in order to identify which drivers are most important in explaining shifts in SOM composition with soil development. Based on previous studies, we hypothesized that absorbance peak areas related to simple plant-derived OM (S-POM) would decrease with increasing soil formation as minerals become increasingly weathered and the formation of OMAs are reduced, hence decreasing the protection of S-POM compounds from microbial degradation and transformation. The peak areas of complex plant-derived OM (C-POM) were expected to increase with soil age due to preferential association with pedogenic iron oxides (Hall et al., 2016; Zhao et al., 2016; Huang et al., 2019). Finally, MOM was expected to strongly increase with soil age

since microbial uptake of OM and the subsequent stabilization in soils leads to an increasing share of microbial-derived SOM (Cotrufo et al., 2013; Cotrufo et al., 2015).

**Reviewer#2 Comment#4 (l. 75-85):**

I can see that in this section a kind of stepwise approach has been explained. Hence, in that respect I would like to suggest to consider making a methodological flowchart and use that in the M&M section (see also my comment just above this one).

**Our response:** We would like to thank the reviewer for encouraging a more comprehensible description of our stepwise methodological approach. However, since there are essentially only two steps, we decided to restate our stepwise approach in the M&M section instead of adding another conceptual figure. Please find here our suggested insertion after the header "2 Materials and methods":

"The present study links information on soil mineralogy and texture (Doetterl et al. 2018) with proxies for microbial transformation and SOM composition to derive a better understanding of long-term SOM dynamics along weathering gradients."

**Reviewer#2 Comment#5 (l. 93):**

You mention climatic condition are homogeneous. Yes, that's true for this particular period in time, but not throughout time. So, as the alluvial deposits are differing in age they have being created under quite different climatic circumstances, and hence, I was wondering whether this fact could have influenced the relative importance of different stabilization mechanisms?

**Our response:** We would like to thank the reviewer for the valuable objection. The authors agree that climatic conditions were not homogeneous over time but subject to change and for instance the deposits on older terraces experienced several glacial-interglacial periods and most likely the climatic changes have led to different weathering rates throughout time. Still, our study focuses on SOM composition changes in relation to more general changes in soil mineralogy along the weathering gradient (following e.g. Harden, 1987; White et al., 1996; White et al., 2005). Hence, in the authors' opinion a more detailed study of how the glacial/interglacial periods affected mineral weathering rates at specific terraces would be beyond the scope of our paper. Therefore, we propose to rephrase the sentence as follows:

"The present climatic conditions  across the study region are semiarid with a mean annual temperature of 16.3 °C and a mean annual precipitation of 315 mm (Reheis et al., 2012)."

References:

Harden, J. W.: Soils Developed in Granitic Alluvium near Merced, California, U.S. Geological Survey, Denver, 1987.

White, A. F., Blum, A. E., Schulz, M. S., Bullen, T. D., Harden, J. W., and Peterson, M. L.: Chemical Weathering of a Soil Chronosequence on Granite Alluvium I. Reaction Rates Based on Changes in Soil Mineralogy, Geochim. Cosmochim. Acta, 60, 2533–2550, 1996.

White, A. F., Schulz, M. S., Vivit, D. V., Blum, A. E., Stonestrom, D. A., and Harden, J. W.: Chemical weathering rates of a soil chronosequence on granitic alluvium: III. Hydrochemical evolution and contemporary solute fluxes and rates, Geochim. Cosmochim. Acta, 69, 1975–1996, https://doi.org/10.1016/j.gca.2004.10.003, 2005.

**Reviewer#2 Comment#6 (l. 99-100):**

I think that this sentence needs some rephrasing: what about somethings as follows: "Soil samples were collected in December 2013 from 1 m³ soil pits located within a circular area with a diameter of approximately 40 km in the North of Merced County."

**Our response:** We fully agree with the reviewer and we will rephrase the sentence as kindly suggested by the reviewer.

"Soil samples were collected in December 2013 from 1 m³ soil pits located within a circular area with a diameter of approximately 40 km in the North of Merced County  (see map in Fig. 1)."

**Reviewer#2 Comment#7 (l. 100):**

you make reference to the fact that this are "glacial periods" but I can see that the considered time-window also includes interglacial periods, and as such I would rephrase this as "quaternary"

**Our response:** We thank the reviewer for the comment. We will modify the sentence and use the expression "quaternary periods".

"The terraces are named based on different quaternary periods that led to the alluvial deposition (Harden, 1987)."

**Reviewer#2 Comment#8 (l. 125):**

Can you specify these correction factors?

**Our response:** Following Beuselinck et al. (1998), clay content is underestimated in silty soil samples due to the "platty" geometry of clay minerals. Therefore, the use of a correction factor is encouraged to correct for the overestimated silt fraction (and thus for the underestimated clay fraction). When applying the following correction, the measurements with a Mastersizer 2000 laser diffraction particle size analyser provide a good correlation with the sieve-pipette method:

$$\% \text{ Silt} = 100 - (\% \text{ estimated clay} + \% \text{ estimated sand})$$

To be more specific in our manuscript, we will add the modified phrase on correction factors:

In silty soils, clay is under- and silt is overestimated due to the planar geometry of clay minerals (Beuselinck et al., 1998). Therefore, correction factors (see Eq. 1) were employed that correct the percentage of silt based on previous studies (Beuselinck et al., 1998; Miller and Schaetzl, 2012):

$$\% \text{ silt} = 100 - (\% \text{ estimated clay} + \% \text{ estimated sand}) \qquad (1)$$

References:

Beuselinck, L., Govers, G., Poesen, J., Degraer, G., and Froyen, L.: Grain-size analysis by laser diffractometry: Comparison with the sieve-pipette method, 32, 193–208, https://doi.org/10.1016/S0341-8162(98)00051-4, 1998.

**Comments regarding the reporting of the results:**

**Reviewer#2 Comment#9 (l. 185-221):**

As mentioned in my overall comment (see above), I think that this sections requires considerable rewriting, including a much clearer engagement with the quantitative information (as being presented in various figures and tables). Moreover, in some cases a bigger effort could be made to assess the statistical value of a given statement. For example, you say in L189-190 "a strong increase" but it would have much more value to indicate whether this strong increase is significant (and at what level, e.g. $p < 0.05$, $p < 0.01$, ect…). In that respect, I would also like to encourage the authors to undertake a much bigger effort in terms of providing more statistical based evidences as regards the values presented in table 2. More precisely as "n = 30" I guess you could have also added standard deviations and / or standard error values, which on its turn could be of use for the statistical interpretation of the results.

**Reviewer#2 Comment#14 (l. 214-216):**

In line of my overall comment (and specific comment related to L185-221) I like to iterate that it is important to mention values as being given in various tables and figures, because the engagement with the quantitative measures (and its statistical interpretation) is very important

**Our response:** We completely agree with the reviever and highly appreciate this comment. To incorporate and explicitly mention the measured values increases the engagement with the reader. Furthermore, the reviewer made us aware that the number of replicates in Table 2 were wrong. As we took the measurements from Doetterl et al. (2018), the mineralogy and texture data have n = 10 instead of n = 30 (which only applies for the SOM parameters). Hence, we were not able to perform ANOVAs and calculate standard errors or standard deviations to underscore the observed trends for soil mineralogy and texture data. Consequently, we decided to attenuate our statements regarding increases/decreases for the mineralogical and textural variabels and coined these changes along the chronosequence as *trends* in the results section. We added information on the significance levels to the parts on the SOM parameters and the model fits where actual statistical tests could be performed based on the sample size (n = 30). In the following, we provide a revised version of the results section with the new parts in red:

**3 Results**

**3.1 Changes in soil properties and SOM composition along the chronosequence**

With increasing soil age, the soil mineral matrix was progressively weathered. Weathering and leaching of cations were reflected in downward trends of CEC from 15.2 to 4.5 in 0-10 cm depth (difference between youngest (0.1 kyrs) and oldest (3000 kyrs) terrace) and from 14.2 to 2.1 in 10-30 cm depth (see Table 2). Similarly, soil pH showed tendencies to decrease from 6.8 to 4.7 in 0-10 cm and 6.8 to 4.3 in 10-30 cm depth (see Table 2). Yet, the gradient observed for CEC and soil pH was not found in base saturation ($B_{sat}$) values which did not show any clear pattern. Moreover, $B_{sat}$ showed no clear trend with increasing soil age but was consistently higher in 10-30 cm depth (except for the youngest 0.1 kyrs terrace). We registered increasing trends of $Si_{total}$ from 241.5 g/kg to 393.4 g/kg in 0-10 cm depth and from 280.3 to 389.4 g/kg in 10-30 cm depth with increasing soil age (see Table 2). $Al_{total}$ and $Fe_{total}$ contents showed the opposite trend, decreasing with soil age in both 0-10 and 10-30 cm depth. However, this decreasing trend was not displayed in the fractions of DCB-extractable Al ($Al_{DCB}$) and Fe ($Fe_{DCB}$) (see Table

2). While $Al_{DCB}$ stagnated along the age gradient at values ranging from 0.7 to 1.3 g/kg in 0-10 cm depth and 0.6 to 1.4 g/kg in 10-30 cm depth, $Fe_{DCB}$ showed increasing tendencies in both depths. In 0-10 cm the values increased from 6.2 to 13.6 g/kg and in 10-30 cm depth from 6.1 to 13.5. The larger $Fe_{DCB}$ fraction among the DCB-extracted pedogenic oxides reflects the high potential of Fe to be quantitatively important for SOM binding. The $Fe_{total}$:$Fe_{DCB}$ ratio showed highest values in the soils of the 19 kyrs terrace (7.6 in 0-10 cm and 6.4 in 10-30 cm depth) (see Table 2). The $Al_{total}$:$Al_{DCB}$ ratio was highest in soils of the 295 kyrs terrace with values of 88.5 in 0-10 cm depth and 106.8 in 10-30 cm depth (see Table 2).

Soil texture showed no clear patterns with soil age but between 0-10 and 10-30 cm depth (see Table 2). Except for the youngest terrace, silt and clay fractions were more abundant in 10-30 cm depth, while the proportion of sand was mostly lower relative to the 0-10 cm depth (see Table 2). Overall bulk total C decreased significantly with soil depth ($p < 0.05$), and with soil age in the 0-10 cm depth ($p < 0.05$). However, the decrease was not linear but peaked at 3 kyrs (3.6 kg/m²; see Fig. 3a). Furthermore, the C:N ratio was significantly lower in 10-30 cm depth ($p < 0.05$) and was significantly lower in the 19, 295, and 3000 kyrs old soils compared to the 0.1 and 3 kyrs old soils ($p < 0.05$; see Fig. 3b). In contrast, $\delta^{13}C$ values showed no significant pattern with soil age but increased significantly with soil depth ($p < 0.05$; see Fig. 3c). The $\delta^{15}N$ values increased significantly with soil age ($p < 0.05$) but showed no significant difference from 0-10 to 10-30 cm depth (see Fig. 3d). Similarly, the peak area of S-POM was significantly lower in the 10-30 cm depth ($p < 0.05$) and remained stable with increasing soil age (see Fig. 4a). Conversely, peak areas related to C-POM and MOM significantly increased with increasing age in 0-10 cm and 10-30 cm depth ($p < 0.05$). Moreover, peak areas were significantly higher in 10-30 cm ($p < 0.05$; see Fig. 4b).

**3.2 Link between soil geochemistry and SOM composition**

The DRIFTS peak areas related to S-POM decreased significantly with soil depth while the C-POM and MOM signals rose significantly with increasing soil age and soil depth (see Fig. 4). This raised the question whether there were identifiable soil properties driving the relative changes of S-POM, C-POM, and MOM. Therefore, we used a modelling approach to predict bulk SOM composition using the soil mineralogy and texture parameters which were not affected by multicollinearity (see Methods). In Table 3, the mean model fit (RMSE, MAE, and R²), relative model fit measures (rRMSE, rMAE), and variable importance coefficients are presented based on 100 iterations of the cross-validated results of the linear models with randomly splitted training and control sets (see Methods).

The C:N ratio was best explained by the $Fe_{total}$:$Si_{total}$ ratio which was the most important variable and only significant model parameter ($p < 0.01$; see Table 3). However, the R² showed high variability and was comparatively low in relation to the other models (R²: $0.32 \pm 0.27$). The rRMSE was 17.9 % and the rMAE was 16.1 % indicating a good prediction error compared to the other models (see Table 3). The $\delta^{15}N$ values were most influenced by clay content, the $Fe_{total}$:$Si_{total}$ ratio, and $Al_{DCB}$. However, despite the high model fit, no model parameter was significant. Regarding the R² value, similar results were obtained for the $\delta^{13}C$ values. However, all model parameters were significant ($p < 0.001$). Yet, based on the absolute values of the variable importance coefficients $Fe_{DCB}$, $Fe_{total}$:$Si_{total}$, and clay were more important than $Al_{DCB}$ (see Table 3). The rRMSE and rMAE were slightly higher for $\delta^{15}N$ compared to the model on the C:N ratio (rRMSE: 24.2 %, rMAE: 20.3 %) and low for $\delta^{13}C$ (rRMSE: -2.2 %, rMAE: -2.2 %). Regarding the peak areas related to S-POM, the most important and significant variable was clay ($p < 0.01$; see Table 3). Yet, the explained variance was low and highly variable (R²: $0.39 \pm 0.33$). This is also reflected in a high rRMSE and rMAE of > 100 % (see Table 3). Conversely, the linear models explaining the variance of the peak areas linked to C-POM and MOM had high R² values (R²: $0.78 \pm 0.10$ and $0.57 \pm 0.27$, respectively) indicating a good fit. In both cases, clay content was the only significant model parameter ($p < 0.01$). Despite the high variable importance coefficient of $Fe_{total}$:$Si_{total}$ in the C-POM model, this parameter was not significant (see Table 3). The rRMSE (C-POM: 56.3 %, MOM: 61.7 %) and rMAE (C-POM: 45.8 %, MOM: 55.1 %) were high in both cases.

**Comments on wrong table cross-referencing:**

**Reviewer#2 Comment#10 (l. 195):**

You make reference to table 3, but I think that isn't the correct, because that's the table representing the output of the regressions. Anyway, please check your in-text references to figures / tables throughout the entire text, because I think they aren't always correct.

**Reviewer#2 Comment#11 (l. 196-198):**

I think that this information comes from table 2? If, so please make reference to the corresponding table. Please also check in other parts of the "Results" and "Discussion" sections whether you always make in-text references to the corresponding tables / figures, because in many cases these seem lacking.

**Reviewer#2 Comment#13 (l. 214):**

I guess this should be "table 3" instead of "table 4"? Please check the entire text (see also my related comments above (L195, L196-198)).

**Our response:** We thank the reviewer for spotting these typos. Indeed, the references to the tables were not correct. We will carefully check all the cross-references and in the revised manuscript and provide a new version without wrong references.

**Reviewer#2 Comment#12 (l. 210):**

Did you perform a multicollinearity analysis? (if so can you give the correlation coefficient matrix and explain which kind of correlation coefficient threshold you did consider in order to say variables were too strongly correlated?)

**Our response:** Yes, we performed a multicollinearity analysis based on the variance inflation factor (VIF) generated with the *caret* package in R (Kuhn, 2020). We excluded factors that had a VIF > 5. It is statistically recommended to remove variables that have VIFs above 5 or 10 (James et al., 2013). The VIF is calculated as a measure that determines how close the parameter estimates ($\hat{\beta}$) are when the full model is fitted to the explanatory variable. A ratio is calculated (see eq. below) and results in values ≥ 1. The lowest possible VIF of 1 corresponds to no collinearity at all.

The advantage of controlling multicollinearity in the dataset with VIF is, that interrelated correlations between the variables can be detected in a safe and reproducible way compared to manual selection of uncorrelated variables in a correlation coefficient matrix. As we did not make use of the correlation coefficient matrix itself, we do not show it here. To improve our manuscript based on the reviewer's comment, we will take action and reformulate the respective section in the M & M section to explain our proceeding in depth (l. 182-184).

"VIFs quantify the collinearity of a variable (Fox and Monette, 1992; Fox and Weisberg, 2019). The VIF of each variable is computed as following (James et al., 2013):

$$VIF(\hat{\beta}_j) = \frac{1}{1 - R^2_{X_j|X_{-j}}} \ (3)"$$

References:

James, G., Witten, D., Hastie, T., Tibshirani, R.: An introduction to statistical learning. Springer: New York, Heidelberg, Dordbrecht, London, 2013.

Kuhn, M.: caret: Classification and Regression Training, https://cran.r-project.org/package=caret, 2020.

**Reviewer#2 Comment#15 (l. 219-221):**

I agree that the R2 values can be of use when comparing the models. However, I have my doubts about the usefulness of the RMSE values, because the RMSE values aren't dimensionless and are depending on the value range of the considered variable. Hence, I think that a relatively scaled variant of RMSE could be more useful, e.g. relRMSE or RPD? In addition, I was wondering whether it could make sense to give (beside a measure for random error) also a measure for the (relative) bias (e.g. %BIAS)?

**Our response:** We agree with the reviewer and implemented his suggestion by providing a measure for the relative RMSE (rRMSE) and relative mean absolute error (rMAE). We therefore suggest the following revised version of Table 4 in the manuscript. Furthermore, we added the significance level of each predictor (see Reviewer#2 Comment#18).

**Table 1: Overview of the linear regression models describing SOM composition (C:N ratio, $\delta^{15}N$ and $\delta^{13}C$), and peak areas of functional groups related to simple plant-derived OM (S-POM), complex plant-derived OM (C-POM) and mainly microbially-derived OM (MOM) based on n = 30 observations. Explanatory variables related to the mineral matrix and soil texture were selected ($Al_{DCB}$, $Fe_{DCB}$, $Fe_{total}:Si_{total}$ ratio, clay content). Absolute and relative model fit parameters (root mean squared error (RMSE and rRMSE), mean absolute error (MAE and rMAE), and $R^2$) with standard deviations were computed following 100 iterations of a Monte-Carlo cross-validation. Higher coefficients of variable importance indicate a higher importance of the variable on the regression slope. The significance levels are denoted as p < 0.001 (***), p < 0.01 (**).**

| | C:N ratio | $\delta^{15}N$ | $\delta^{13}C$ | S-POM | C-POM | MOM |
|---|---|---|---|---|---|---|
| **RMSE** | 2.0 ± 0.5 | 1.2 ± 0.5 | 0.6 ± 0.2 | 2.9 ± 1.0 | 3.2 ± 0.7 | 20.3 ± 5.7 |
| **rRMSE** | 17.9 | 24.4 | -2.2 | 128.9 | 56.3 | 61.7 |
| **MAE** | 1.8 ± 0.4 | 1.0 ± 0.4 | 0.6 ± 0.1 | 2.5 ± 0.9 | 2.6 ± 0.7 | 18.1 ± 5.5 |
| **rMAE** | 16.1 | 20.3 | -2.2 | 111.1 | 45.8 | 55.1 |
| **R²** | 0.32 ± 0.27 | 0.64 ± 0.24 | 0.61 ± 0.32 | 0.39 ± 0.33 | 0.78 ± 0.10 | 0.57 ± 0.27 |
| **Coefficients of variable importance** | | | | | | |
| **$Al_{DCB}$** | 1.3 | 1.3 | 3.3 ** | 2.3 | 1.0 | 1.1 |
| **$Fe_{DCB}$** | 1.8 | 0.6 | 5.2 *** | 2.1 | 1.0 | 1.6 |
| **$Fe_{total}:Si_{total}$** | 3.2 ** | 1.4 | 5.0 *** | 1.4 | 5.0 | 0.8 |
| **Clay** | 0.4 | 1.5 | 4.1 *** | 2.2 ** | 2.8 ** | 3.7 ** |

**Reviewer#2 Comment#16 (l. 234-236):**

Is this significant?

**Our response:** We thank the reviewer for the comment and will modify the respective lines as following in red:

"SOM was increasingly processed as indicated by a significant decrease of the C:N ratio and $\delta^{15}N$ over time and decreases of $\delta^{13}C$ with soil depth. These developments were accompanied by significant increases of the peak

areas related to complex plant-derived OM (C-POM) and microbial derived OM (MOM), and constant peak areas related to simple plant-derived OM (S-POM) (see Fig. 5)."

**Reviewer#2 Comment#17 (l. 248-250):**

I think this interpretation should be made with more care, because as the vegetation type did vary over time (as a function of climate variations), the quantity of C input as well as associated origin and stability will have been different along you chronosequence (see also my comment related to L39). Hence, I think some more critical reflection is required here when interpreting the results.

**Our response:** We agree with the reviewer that the vegetation type varied over time (and thus the quantity and quality of C input) as a function of climate variations. However, this was not reflected in the SOM pools studied here (i.e. they were too coarse to detect more minor differences). Therefore, as this was not the main point of our study (as such as assessment would also require a different analytical approach), we believe that further discussion of this topic is beyond the scope of the present paper. We propose to account for the influence of vegetation type and climate throughout the quaternary in the following sentence:

"Throughout the quaternary, climate and vegetation were variable. Still, S-POM did not decrease along the chronosequence indicating a  steady supply by above- (light fraction of litter compounds) and belowground input (low-molecular weight root exudates, Nardi et al., 2005) particularly in the topsoil layer."

**Reviewer#2 Comment#18 (l. 274-276):**

When looking to the coefficients of variable importance in table 3, I can see that Fetotal:Sitotal has a value of 3.2 whereas that of FeDCB is 1.8. Hence, following your statement, I understand that 1.8 is considered as not significant? I that correct? Can you please explain to the readers somewhere (e.g. in the "material and methods" section) how to interpret these values in terms of their statistical meaning?

**Our response:** We thank the reviewer for the comment and agree that these statistics were not described clearly enough. Our assessment of variable importance is based on absolute t-values which are calculated by dividing the parameter estimate ($\beta_i$) by the standard error of estimation ($SE_i$) (James et al., 2013). Hence, the t-value is a measure to determine the effect that each parameter makes on the regression slope (Grömping, 2015). To give information on the significance of the impact, we added the p-value attributed to each predictor (Bring, 1996). We propose to add the following section in the M&M section:

" To assess the variable importance of the model parameters we used the absolute t-values which are calculated by dividing the parameter estimate ($\beta_i$) by the standard error of estimation ($SE_i$) (James et al., 2013; Kuhn, 2020). The t-value is a measure to determine the effect that each parameter makes on the regression slope (Grömping, 2015). To give information on the significance of the impact, we added the p-value attributed to each predictor (Bring, 1996)."

References:

Bring, J.: A geometric approach to compare variables in a regression model. The American Statistician, 50, 57-62, 1996.

Grömping, U.: Variable importance in regression models. WIREs Comput Stat, 7, 137-152. https://doi.org/10.1002/wics.1346, 2015.

James, G., Witten, D., Hastie, T., Tibshirani, R.: An introduction to statistical learning. Springer: New York, Heidelberg, Dordbrecht, London, 2013.

Kuhn, M.: caret: Classification and Regression Training, https://cran.r-project.org/package=caret, 2020.

**Reviewer#2 Comment#19 (Fig. 1):**

Can you add an extra subplot to this figure indicating where this study site is located within the western part if the USA / California?

**Our response:** We agree with the author that the spatial context becomes clearer to the reader when adding a map of the larger spatial context. Please find below the new map:

[Figure]

**Reviewer#2 Comment#20 (Table 2):**

This table has multiple layout issues. First of all, I think "30" is missing in the last 5 rows of the first column (so "10-" should be "10-30"). Moreover, I think that it could be a better idea to switch rows with columns because 17 columns re fare too much (I can't read the headings properly) and/or this figure should be made in landscape format. Another suggestion could be to split the table in 2 sub-tables (or 2 separate tables) one considering 0-10 and the other considering 10-30. Finally, I was also wondering why you couldn't present this information in graphs (with "soil age" on the x-ax) just as you did in figure 3 and 4? And last but not least, I like to reiterate my comment about providing more statistical measures, because as "n = 30", I guess you could have also added standard deviations and / or standard error values. (which you actually do in figures 3 and 4).

**Our response:** We agree with the author and tackled the multiple layout issues that were kindly listed by the reviewer. To correctly display the 17 columns and improve readability, we followed the reviewer's suggestion and opted for a landscape format (see below). We think that the information is best presented in a table due to the sheer amount of variables. To show only the target variables in figures and condense the supporting large amount of information in a table increases the consistency of our manuscript.

The geochemical parameters were measured by Doetterl et al. (2018) and by mistake we stated a sample size of n = 30 for these measurements. However, no replicates were measured but selectively duplicates. Hence, we are not able to provide nor an estimation of standard errors or standard deviation nor an indication of significant differences between the terraces and/or soil depth for the mineralogy and texture data since the sample size totals n = 10. Following Comment#9 and #14, we addressed this limitation in the results section by describing the changes in mineralogy and texture as trends.

Please find here the revised table:

**Table 2:** Overview of soil mineralogy and texture changes with increasing soil age in 0-10 and 10-30 cm depth (n = 10). The mineralogy was measured using inductively coupled plasma-atomic emission spectrometry (ICP-AES). Texture was quantified with a laser diffraction particle size analyser. Pedogenic oxides were extracted with dithionite-citrate-bicarbonate (DCB).

| Soil depth [cm] | Soil age [kyrs] | $Al_{total}$ [g/kg] | $Fe_{total}$ [g/kg] | $Si_{total}$ [g/kg] | $Fe_{total}:Si_{total}$ | $Al_{DCB}$ [g/kg] | $Fe_{DCB}$ [g/kg] | $Fe_{total}:Fe_{DCB}$ | $Al_{total}:Al_{DCB}$ | Ti:Zr | Clay [%] | Silt [%] | Sand [%] | CEC [meq/100g] | $B_{sat}$ [%] | pH |
|---|---|---|---|---|---|---|---|---|---|---|---|---|---|---|---|---|
| 0-10 | 0.1 | 82.7 | 36.5 | 241.5 | 0.15 | 1.3 | 6.2 | 5.9 | 65.1 | 21.68 | 15.2 | 58.4 | 26.4 | 15.2 | 86.6 | 6.8 |
| 0-10 | 3 | 65.0 | 35.3 | 293.1 | 0.12 | 1.0 | 9.5 | 3.7 | 66.2 | 19.62 | 9.1 | 62.6 | 28.2 | 13.4 | 109.4 | 6.6 |
| 0-10 | 19 | 81.1 | 38.4 | 267.2 | 0.14 | 1.1 | 6.6 | 7.6 | 86.9 | 26.85 | 11.2 | 64.3 | 24.5 | 24.7 | 81.1 | 5.9 |
| 0-10 | 295 | 63.9 | 18.2 | 338.6 | 0.05 | 0.7 | 6.2 | 2.9 | 88.5 | 13.82 | 13.7 | 50.3 | 36.0 | 4.6 | 103.7 | 6.1 |
| 0-10 | 3000 | 29.3 | 23.1 | 393.4 | 0.06 | 1.3 | 13.6 | 1.7 | 22.3 | 10.86 | 11.1 | 68.1 | 20.8 | 4.5 | 58.3 | 4.7 |
| 10-30 | 0.1 | 92.2 | 37.1 | 280.3 | 0.13 | 1.0 | 6.1 | 6.1 | 102.8 | 18.07 | 4.9 | 63.1 | 32.0 | 14.2 | 74.3 | 6.8 |
| 10-30 | 3 | 73.4 | 39.6 | 306.2 | 0.13 | 0.9 | 10.4 | 3.8 | 80.3 | 14.22 | 12.9 | 69.2 | 17.9 | 9.9 | 116.6 | 6.2 |
| 10-30 | 19 | 87.1 | 43.7 | 273.0 | 0.16 | 1.3 | 7.2 | 6.4 | 84.7 | 24.42 | 17.4 | 69.3 | 13.4 | 20.8 | 95.5 | 6.0 |
| 10-30 | 295 | 64.9 | 18.4 | 348.0 | 0.05 | 0.6 | 6.5 | 2.8 | 106.8 | 14.60 | 19.0 | 53.3 | 27.7 | 2.1 | 131.2 | 5.8 |
| 10-30 | 3000 | 36.7 | 25.3 | 389.4 | 0.06 | 1.4 | 13.5 | 1.9 | 26.2 | 10.13 | 18.3 | 69.2 | 12.5 | 2.1 | 85.3 | 4.3 |

**Reviewer#2 Comment#21 (Fig. 3 & 4):**

I would like to suggest to use a logarithmic x-ax in order to place the soil ages accordingly on it (instead of using just equal distances between the different soil ages, and hence, not having 'a real numerical x-axe').

**Our response:** We agree with the reviewer and thank for the suggestion. Kindly find below the revised figures with logarithmic x-axes.

[Figure]

---

## Author Response (AR1)

Biogeosciences Discuss., referee comment RC1
https://doi.org/10.5194/bg-2021-295-RC1, 2021

[Figure]

**Comment on bg-2021-295**

Xavier Dupla (Referee)
* * *
Referee comment on "Soil geochemistry as a driver of soil organic matter composition: insights from a soil chronosequence" by Moritz Mainka et al., Biogeosciences Discuss., https://doi.org/10.5194/bg-2021-295-RC1, 2021
* * *
An overall excellent paper. Informative, concise and very well written. I am confident that the scientific community will welcome it warmly. Please find, some minor comments below:

line 58: remove thereby which is misleading with the previous sentence

line 61: oxide do not "become" positively charged at low pH values, they are positively charged on the whole pH range of almost all soils (check pzc values). If you want to open the pandora box of variable charges, it is difficult to speak about the protonation of surfaces without saying that several OM functional groups too protonates when pH becomes acidic. Furthermore, if you want to maintain this sentence about acidic soils, then you could more explicitly mention that soil acidification is a key process behind soil weathering.

line 79: this section contradicts what you say from line 47 onwards. Rephrasing either the upper section (lines 47-52) of this one (lines 79-85) might help

Discussion section: overall excellent. However, you did not notice any significant decrease in base saturation along your chrono-sequence which contradicts general description of soil weathering sequences. This aspect is extremely interesting and should be discussed.

Non-binding suggestion : your discussion sticks very closely to the parameters. I was expecting your paper to zoom out at some point in order to 1) discuss how the climate and geomorphological changes that happened in your 3 million-year sequence may have impacted your results 2) discuss the general impact of your findings on our understanding of soil weathering, 2) outline the limits of your study and what should be done to go further.

[Figure]

Biogeosciences Discuss., referee comment RC2
https://doi.org/10.5194/bg-2021-295-RC2, 2022

[Figure]

**Comment on bg-2021-295**

Anonymous Referee #2
* * *
Referee comment on "Soil geochemistry as a driver of soil organic matter composition: insights from a soil chronosequence" by Moritz Mainka et al., Biogeosciences Discuss., https://doi.org/10.5194/bg-2021-295-RC2, 2022
* * *
**General Appreciation / Overall Comment:**

This is an interesting paper addressing a hot topic in the domain of SOC research (i.e. understanding long-term SOC stabilization / decomposition mechanisms). The paper is well written based on the innovative research idea of considering a soil chronosequence and reveals that way some new insights. However, the results section has been worked out rather weakly, in that sense that it is rather brief and descriptive, but more critically, there is a lack of making use of the quantitative measures / evidences (as being represented by the various figures in tables and figures) as well as associated statistical interpretation. Hence, in particular, this section requires extra attention when improving the manuscript based on following more specific comments:

**Specific Comments:**

L 36: You mention "responds to land use change" but I'm not too sure whether it is that relevant to mention it here, because land use change will have an impact on SOC stabilization mechanisms on a much shorter time period (scale of 10-100 years), whereas you are looking to a timescales exceeding 1000s of years.

L 40 – 65 I like the idea of the interaction between physical and chemical stabilization mechanisms and associated changes over time. But as regards the physical protection, I can see that the focus is mainly on sorption / binding of OM to mineral surfaces (and as such also the importance of clay % and type of clay in this context). However, I was wondering whether soil aggregate formation (macros and micros) shouldn't be considered as well / more explicitly here? (or is this not the right time scale. But if so, I guess you should neither mention 'land use change' – see my comment just above this one).

L66-69 & L 75-85: I think that the information given in these sections mainly belongs to the Material & Methods section. Hence, I would like to suggest to rewrite the end of the introduction section so that you end with making the problem statement clear followed by your main objective(s), i.e. the contribution of present research in addressing this particular problem statement.

L 75-85 I can see that in this section a kind of stepwise approach has been explained. Hence, in that respect I would like to suggest to consider making a methodological

flowchart and use that in the M&M section (see also my comment just above this one)

L 93 You mention climatic condition are homogeneous. Yes, that's true for this particular period in time, but not throughout time. So, as the alluvial deposits are differing in age they have being created under quite different climatic circumstances, and hence, I was wondering whether this fact could have influenced the relative importance of different stabilization mechanisms?

L 99 – 100 I think that this sentence needs some rephrasing: what about somethings as follows: "Soil samples were collected in December 2013 from 1 m3 soil pits located within a circular area with a diameter of approximately 40 km in the North of Merced County."

L 100 you make reference to the fact that this are "glacial periods" but I can see that the considered time-window also includes interglacial periods, and as such I would rephrase this as "quaternary".

L125 Can you specify these correction factors?

L 185-221 As mentioned in my overall comment (see above), I think that this sections requires considerable rewriting, including a much clearer engagement with the quantitative information (as being presented in various figures and tables). Moreover, in some cases a bigger effort could be made to assess the statistical value of a given statement. For example, you say in L189-190 "a strong increase" but it would have much more value to indicate whether this strong increase is significant (and at what level, e.g. p < 0.05, p < 0.01, ect…). In that respect, I would also like to encourage the authors to undertake a much bigger effort in terms of providing more statistical based evidences as regards the values presented in table 2. More precisely as "n = 30" I guess you could have also added standard deviations and / or standard error values, which on its turn could be of use for the statistical interpretation of the results.

L 195 You make reference to table 3, but I think that isn't the correct, because that's the table representing the output of the regressions. Anyway, please check your in-text references to figures / tables throughout the entire text, because I think they aren't always correct.

L 196-198 I think that this information comes from table 2? If, so please make reference to the corresponding table. Please also check in other parts of the "Results" and "Discussion" sections whether you always make in-text references to the corresponding tables / figures, because in many cases these seem lacking.

L 210 Did you perform a multicollinearity analysis? (if so can you give the correlation coefficient matrix and explain which kind of correlation coefficient threshold you did consider in order to say variables were too strongly correlated?).

L 214 I guess this should be "table 3" instead of "table 4"? Please check the entire text (see also my related comments above (L195, L196-198)).

L214-216 In line of my overall comment (and specific comment related to L185-221) I like to iterate that it is important to mention values as being given in various tables and figures, because the engagement with the quantitative measures (and its statistical interpretation) is very important.

L219-221. I agree that the R2 values can be of use when comparing the models. However, I have my doubts about the usefulness of the RMSE values, because the RMSE values aren't dimensionless and are depending on the value range of the considered variable. Hence, I think that a relatively scaled variant of RMSE could be more useful, e.g. relRMSE

or RPD? In addition, I was wondering whether it could make sense to give (beside a measure for random error) also a measure for the (relative) bias (e.g. %BIAS)?

L234-236 Is this significant?

L 248-250 I think this interpretation should be made with more care, because as the vegetation type did vary over time (as a function of climate variations), the quantity of C-input as well as associated origin and stability will have been different along you chronosequence (see also my comment related to L39). Hence, I think some more critical reflection is required here when interpreting the results.

L 274 – 276 When looking to the coefficients of variable importance in table 3, I can see that Fetotal:Sitotal has a value of 3.2 whereas that of FeDCB is 1.8. Hence, following your statement, I understand that 1.8 is considered as not significant? I that correct? Can you please explain to the readers somewhere (e.g. in the "material and methods" section) how to interpret these values in terms of their statistical meaning?

Figure 1: Can you add an extra subplot to this figure indicating where this study site is located within the western part if the USA / California?

Table 2: This table has multiple layout issues. First of all, I think "30" is missing in the last 5 rows of the first column (so "10-" should be "10-30"). Moreover, I think that it could be a better idea to switch rows with columns because 17 columns re fare too much (I can't read the headings properly) and/or this figure should be made in landscape format. Another suggestion could be to split the table in 2 sub-tables (or 2 separate tables) one considering 0-10 and the other considering 10-30. Finally, I was also wondering why you couldn't present this information in graphs (with "soil age" on the x-ax) just as you did in figure 3 and 4? And last but not least, I like to reiterate my comment about providing more statistical measures, because as "n = 30", I guess you could have also added standard deviations and / or standard error values. (which you actually do in figures 3 and 4).

Figure 3 & 4: I would like to suggest to use a logarithmic x-ax in order to place the soil ages accordingly on it (instead of using just equal distances between the different soil ages, and hence, not having 'a real numerical x-axe').